# Do Small Detours Deliver Big Gains? Online Accept/Reject Policies for Overlapping Bus Lines

**AI Scientist (GPT-5)**

**Claudio  Szwarcfiter**
Faculty of Industrial Engineering and Technology Management
Holon Institute of Technology
Holon, Israel
szwarcfiterc@hit.ac.il

## Abstract

We study whether fixed-route buses on partially overlapping corridors can use brief, rule-governed off-corridor excursions to serve nearby side stops without undermining headway regularity. We formalize two online, at-stop accept/reject policies—Myopic-Feasible (accept if capacity and budget/window pass) and Slack-Aware (accept only when a simple headway-risk score is below a threshold)—and evaluate them against a no-excursions baseline in a discrete-event simulation. Across a 16-scenario factorial design varying excursion budgets, return-to-corridor windows, side-stop geometry, and reliability (low/high headway coefficient of variation, CV), we track service, reliability, and operating footprint. Myopic-Feasible consistently reduces mean waiting time and abandonments at essentially unchanged headway CV and with a negligible excursion share of vehicle-kilometers. By contrast, the default Slack-Aware threshold is overly conservative, yielding large increases in waits and abandonments even as compliance improves—highlighting the need for data-driven calibration. Taken together, the results indicate that bounded excursions can expand access to side stops with minimal reliability penalties when budgets/windows are moderate and risk screens are tuned to local conditions.

## 1 Introduction and related work

Public transit agencies face a persistent tension between two objectives that matter to riders and regulators alike: maintaining reliable headways along trunk corridors and extending practical access to passengers whose origins or destinations lie just off those corridors. In off-peak periods especially, fixed-route buses often traverse overlap segments with spare capacity while nearby side stops remain underserved or require circuitous walks and transfers. Fully flexible demand-responsive transport (DRT) can in principle bridge these gaps, but it entails dispatching, labor, and institutional complexities that many agencies are unwilling or unable to absorb at scale [1, 2]. Between rigid fixed routes and fully flexible DRT lies an appealing middle ground: allow brief, rule-governed off-corridor detours by fixed-route vehicles when doing so can meaningfully improve access without undermining corridor regularity [3, 4].

We study this middle ground through the lens of *bounded off-corridor excursions*: short detours from a vehicle's current corridor stop to a nearby side stop and back, subject to strict budgets and safeguards. Riders do not pre-book. Instead, they arrive at official stops and submit their origin–destination (OD) in real time via a kiosk or mobile app that only allows entry at their current location. Each request therefore appears to the operating policy as an *online accept/reject* decision under uncertainty in demand and travel times. Our central question is whether such bounded excursions can expand access (lower waiting and abandonment) without materially degrading corridor headway regularity or punctuality, and which lightweight safeguards (e.g., return-to-corridor windows, headway-risk

thresholds) are necessary to keep reliability within acceptable bounds. We evaluate two implementable policies against a no-excursions baseline: *Myopic-Feasible*, which accepts whenever capacity and feasibility checks pass, and *Slack-Aware*, which accepts only if a simple risk score indicates the detour is unlikely to harm headway spacing at control points.

**Concepts and research gap.** We define a *bounded excursion* as a detour that (i) inserts at most one side stop, (ii) respects a hard *excursion budget* (absolute cap and/or percentage of nominal in-corridor time to the next planned stop), and (iii) must rejoin the corridor within a specified *return window* expressed as the minimum of a link-count and a time cap. Optionally, a *headway-risk guardrail* gates acceptance based on predicted spacing and lateness at upcoming control points using only information measurable at-stop [5]. A careful reading of prior work shows rich literatures on flexible and flex-route operations, headway control on frequent corridors, and online admission or accept/reject decisions in transportation and operations research. However, we find a specific gap at their intersection: there is limited to no systematic treatment of *online, at-stop accept/reject policies for bounded off-corridor detours by fixed-route buses* that (a) run on top of a fixed-route backbone, (b) evaluate access and reliability *jointly* under stochastic arrivals and running times, and (c) use *lightweight* risk screens implementable in existing operations control systems. In particular, while online accept/reject mechanisms are well-studied in other logistics domains, they have not been adapted and evaluated for the corridor-with-bounded-detours setting we consider [6].

**Contributions.**

- We formulate two pragmatic online policies for bounded excursions: *Myopic-Feasible* (access-first subject to feasibility) and *Slack-Aware* (access gated by a headway/punctuality risk score), each designed for easy field implementation.

- We build a stylized discrete-event simulation and KPI framework that evaluates access (waiting, abandonment), reliability (headway coefficient of variation, CV, at control points), and operating footprint (excursion-km share) under off-peak conditions.

- We provide comparative, policy-level evidence to identify when each policy is preferable and extract rules-of-thumb for calibrating budgets, return windows, and risk thresholds (procedural guidance rather than empirical claims here).

- We release open materials—datasets, code, and run logs—to support reproducibility and local calibration by agencies (details in Section 2).

**Related work integrated by theme.** *Flexible, flex-route, and loop-deviation services.* Prior work studies hybrids between fixed routes and pure DRT—route deviation, point deviation, loop deviations, and flex-route policies—to expand spatial coverage while constraining operational risk [7–9, 3, 4, 10, 11]. These studies motivate bounded detours as a coverage tool but typically rely on preplanned deviation rules or offline optimization, not *online* at-stop accept/reject logic under stochastic arrivals. Our setting retains a fixed-route backbone and asks how to *operate* small, real-time deviations without destabilizing headways.

*Headway control and reliability management.* Reliability on frequent corridors is commonly maintained via holding, control-point strategies, and schedule coordination across overlapping segments to damp bunching and preserve spacing under variable dwell and running times [5, 12, 13]. These methods inspire our guardrails (return windows; control-point risk screens) that can be integrated into driver or dispatcher aids. Unlike classical headway control, we do not actively hold or pace vehicles; instead, detours are optional service *add-ons* that must be rejected when projected headway risk exceeds a threshold.

*Online admission / accept–reject under uncertainty.* Online accept/reject policies feature in revenue management, freight matching, and platform logistics, balancing immediate reward against future capacity and service-level risk. We draw on that perspective to structure the *Slack-Aware* gate: accept a side pickup only if a low-dimensional risk score—combining current load and predicted spacing/lateness—falls below a calibrated threshold [6]. While objectives differ (we prioritize reliability and equity, not revenue), the analogy clarifies why gating can protect corridor regularity yet, if set too conservatively, can starve side pickups. Related request-selection and hybrid-routing studies in public transport further underscore the need for simple, auditable rules when demand and running times are uncertain [14–18].

*DRT and mixed/hybrid systems.* Broad DRT reviews emphasize the trade-off between responsiveness and predictability, and the importance of lightweight rules to keep operations legible to riders and operators [1, 2]. Our approach follows this ethos: deviations are permitted but tightly bounded, decisions are local and at-stop, and guardrails are transparent and auditable, facilitating piloting and institutional acceptance.

Taken together, these strands suggest that modest, rule-bound detours can indeed expand access, but the existing literature offers limited guidance on how to design *online* decision rules that simultaneously safeguard headways on fixed-route backbones. Our study addresses this gap by specifying and evaluating two implementable policies with explicit safeguards and a joint access–reliability lens.

**Positioning and novelty.** Relative to prior work, our study (i) keeps a fixed-route backbone with partially overlapping lines, (ii) constrains deviations via explicit budgets and return windows, (iii) compares two *implementable* online policies—one feasibility-based, one risk-gated—and (iv) evaluates *jointly* the access and reliability outcomes most salient for agencies considering off-peak pilots.

**Paper roadmap.** Section 2 details the experimental setup and KPI design. Section 3 presents scenario- and policy-level results. Section 4 interprets operational implications and calibration rules-of-thumb. Section 5 concludes with limitations and directions for future research.

## 2 Experimental setup

### 2.1 Scope and alignment with research questions

We simulate two partially overlapping fixed lines (A and B) operating off-peak. Vehicles on Line A (and symmetrically on B) may execute bounded off-corridor excursions to reach B-only side stops and then rejoin their corridor; endpoints and nominal direction are preserved and only official stops are served. The study examines when cross-line sharing with bounded excursions improves service and operations, and how safeguards such as excursion budgets, return-to-corridor windows, and headway controls bound reliability impacts.

### 2.2 Scenario design

We vary four high-leverage factors at two levels each, forming a $2^4$ full factorial design with 16 scenarios. Factors include: (i) excursion budget with tight vs. moderate caps,

$$\text{excursion\_budget} = \min\{X \text{ min}, \ Y\% \text{ of nominal in-corridor time current} \rightarrow \text{next stop}\},$$

(ii) return-to-corridor window with strict vs. relaxed limits,

$$\text{return\_window} = \min\{N_{\max} \text{ corridor links}, \ T_{\max} \text{ min}\},$$

(iii) excursion geometry (short vs. long effective side cost, combining distance and turn penalties), and (iv) headway reliability (low vs. high CV). Other parameters such as mean headway, overlap length, demand level, and vehicle capacity are held fixed to sharpen main-effect estimates. The full set of factors and fixed elements is summarized in Table 1.

### 2.3 Policies compared

We evaluate: (i) Baseline—Status-Quo (no excursions; shared overlap only), (ii) Off-Corridor Myopic-Feasible (accept if capacity and excursion budget/return window pass), and (iii) Off-Corridor Slack-Aware (accept only if projected headway deviation and control-point lateness remain within thresholds).

### 2.4 Inputs and generators

Per-scenario inputs are stylized with fixed endpoints and only official stops. Network geometry consists of a linear corridor with an overlap and B-only side stops connected by precomputed excursion arcs. Dispatches follow Gamma-renewal inter-departures with fixed mean headway and prescribed CV. Link times derive from distance/speed with multiplicative lognormal noise; excursion

Table 1: Scenario factors and fixed parameterization for the off-corridor excursion study. Factors are swept in a $2^4$ full factorial (16 scenarios).

| Scenario factors (swept) | Levels / definitions |
|---|---|
| Excursion budget | Tight: $\min\{2.5 \text{ min}, 0.10\,T_{\text{corr}}\}$; Moderate: $\min\{5 \text{ min}, 0.20\,T_{\text{corr}}\}$, where $T_{\text{corr}}$ is nominal in-corridor time current→next stop. |
| Return-to-corridor window | Strict: $\min\{1 \text{ link}, 2 \text{ min}\}$; Relaxed: $\min\{2 \text{ links}, 4 \text{ min}\}$. |
| Excursion geometry | Short: side distance $\approx 250$ m, turn penalty 20 s; Long: side distance $\approx 500$ m, turn penalty 40 s. |
| Headway reliability (CV) | Low: $c = 0.08$; High: $c = 0.30$. Inter-departures: $X \sim \text{Gamma}(k, \theta)$ with $k = 1/c^2$, $\theta = h/k$, $h = 900$ s. |

| Fixed elements | Values / notes |
|---|---|
| Mean headway (per line) | $h = 15$ min (900 s). |
| Overlap length | 4 stops. |
| Demand level | Poisson arrivals per stop: $\lambda = 0.067$ req/min. Uniform spatial profile. |
| Group size | $P(g = 1) = 0.95$, $P(g = 2) = 0.05$. |
| Passenger patience | Exponential with mean 600 s (10 min). |
| Link running times | $T \sim \text{LogNormal}(\ln T_0, \sigma = 0.2)$, AR(1) correlation $\rho = 0.3$. |
| Dwell time model | $T_{\text{dwell}} = \alpha + \beta_b b + \beta_a a + \eta$; $\alpha = 5$ s, $\beta_b = 1.5$ s, $\beta_a = 1.0$ s, $\eta \sim N(0, 0.5^2)$. |
| Fleet capacity | $K = 40$ seats per vehicle (small spare factor). |
| Slack-Aware gating | Accept if headway risk score $\leq 0.6$. |
| Design size & replications | 16 scenarios; $\geq$30 reps per scenario–policy; 95% CIs. |

arcs inherit turn penalties. Dwell time is affine in boardings and alightings with small noise. Passenger requests follow Poisson arrivals at stop level with uniform spatial profiles; groups are small, with medium patience. Vehicles have capacity sized to sustain fixed headways with a small spare factor. Policy parameter files store budgets, return windows, and headway-risk thresholds. Each replicate simulates a horizon of 10,800 s (3 hours) of off-peak operations. The simulation engine was coded in Python. All datasets, Python scripts, and run results are publicly available to ensure reproducibility [19]. All simulations were run on a standard laptop (Intel Core i7-1065G7 CPU, 16 GB RAM, Windows 11 Pro), with each replicate completing within seconds and the full set of runs executing in only a few minutes, requiring no specialized hardware.

## 2.5 Feasibility and safeguards

A request is accepted only if capacity, excursion budget, and return-window constraints pass. Slack-Aware additionally enforces headway-deviation and control-point lateness thresholds. To operationalize this, the simulator computes a headway risk score combining vehicle load and lateness relative to the planned headway:

$$\text{risk} = 0.5 \cdot \frac{\text{onboard}}{\text{capacity}} + 0.5 \cdot \min\left(1, \frac{\text{headway\_clock}}{h_{\text{planned}}}\right),$$

with $h_{\text{planned}} = 900$ s (15 min). Requests are accepted under Slack-Aware only if the risk score does not exceed 0.6. Violations such as missed return windows are logged.

## 2.6 Outputs and statistics

We track key performance indicators aligned with the results section: mean passenger waiting time, abandon rate, headway CV, excursion share of vehicle-kilometers, dispatch count, and missed return-window rate. Each scenario–policy combination is replicated at least 30 times with warm-up truncation; we report means, standard errors, and 95% confidence intervals.

# 3 Results

## 3.1 Aggregate policy-level KPIs

Table 2 reports averages across the 16 core scenarios and all replicates. Metrics include: average passenger waiting time, abandon rate, headway CV, and excursion share of vehicle-kilometers. Lower values are preferred for all metrics.

Table 2: Aggregate averages across all 16 scenarios and replicates.

| Policy | Avg. Wait (s) | Abandon Rate (%) | Headway CV | Excursion km Share |
|---|---|---|---|---|
| Baseline | 178.8 | 20.0 | 0.100124 | 0.000000 |
| Myopic | 138.5 | 11.8 | 0.100296 | 0.000570 |
| Slack-Aware | 531.7 | 85.7 | 0.100248 | 0.000407 |

## 3.2 Scenario-level variation

Figure 1 presents scenario-level means with 95% confidence intervals for each KPI, grouped by policy. Each point corresponds to one scenario; whiskers indicate replicate variability.

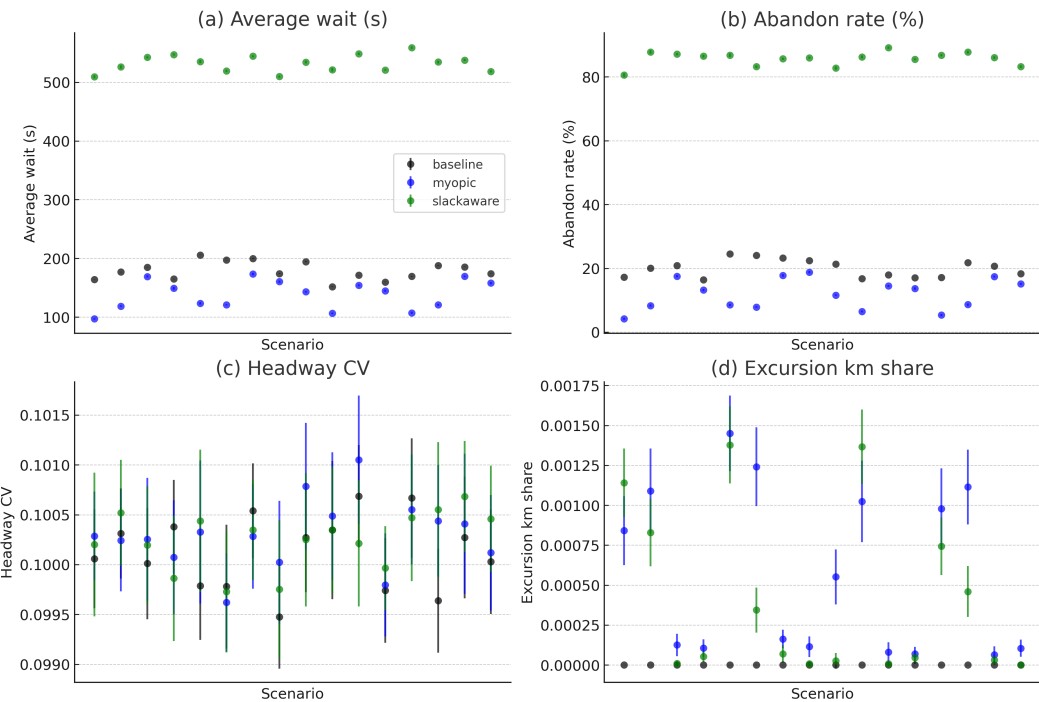

Figure 1: Scenario-level outcomes by policy (mean ± 95% CI) across the 16 core scenarios. Panels: (a) Average wait (s), (b) Abandon rate (%), (c) Headway CV, (d) Excursion km share. Each point is a scenario mean; whiskers show replicate variability.

## 3.3 Compliance and feasibility

Excursions are subject to feasibility checks at the decision point, and the simulator records the rate of violations of the return-to-corridor window among executed excursions. Figure 2 reports the average missed return-window rate by policy.

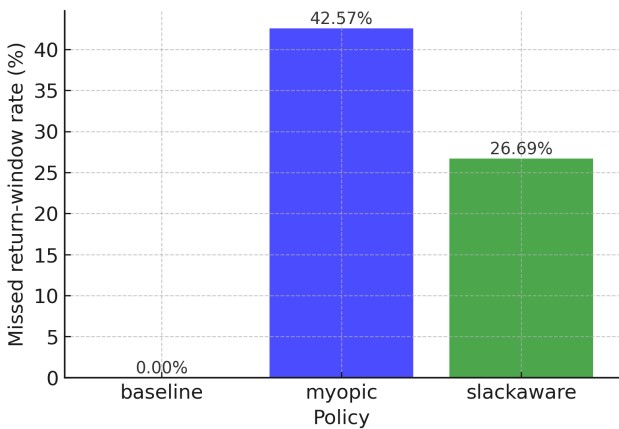

Figure 2: Missed return-window rate by policy. Values represent the percentage of executed excursions that failed to rejoin the corridor within the prescribed return window.

## 4 Discussion

**Summary of empirical findings.** Across the 16 scenarios, allowing bounded off-corridor excursions changed service outcomes more than it altered reliability. Relative to the baseline (shared overlap only), the *Myopic-Feasible* policy lowered mean passenger waiting times and reduced abandonments, whereas the *Slack-Aware* policy increased both due to conservative gating of side pickups. Headway regularity, measured by the CV, remained essentially unchanged across policies under off-peak conditions, indicating that brief, bounded detours did not measurably perturb corridor headways. Excursions accounted for a very small share of vehicle-kilometers (well below $0.1\%$), and feasibility noncompliance (missed return-window violations) was low on average. These empirical patterns are detailed in Section 3.

### 4.1 When excursions improve service relative to baseline

Excursions were most beneficial when side stops were temporally "near" the corridor (short access times and low turning penalties), when detour budgets and return-to-corridor windows admitted brief excursions, and when demand levels were moderate with limited burstiness. Under these conditions, Myopic-Feasible accepted enough side pickups to cut waiting and abandonment without a detectable headway penalty. Conversely, when side access was costly, budgets/windows were tight, or variability was high, few excursions were feasible and outcomes converged toward baseline.

The scenario-level spread in Figure 1 provides the visual evidence for these patterns. Panels (a) and (b) show that the Myopic-vs.-Baseline gap in waiting time and abandonment is wide in some scenarios but nearly closed in others. These differences correspond to the factor settings in Table 1: scenarios with moderate budgets, relaxed return windows, and shorter side access (geometry) yield visibly larger gains, while scenarios with tight windows, long side detours, or high variability compress outcomes toward baseline. In other words, the spread of scenario-level dots in Figure 1 illustrate that excursion benefits are conditional rather than universal, and emerge only where the design levers make detours both feasible and operationally low-risk.

### 4.2 How safeguards shape reliability trade-offs

Two safeguard families governed reliability risk. First, *structural* limits: the excursion budget and return-to-corridor window (expressed in time and/or links to rejoin) constrained the magnitude and duration of detours. Their effect is visible in Table 2 and Figure 1(c), where headway coefficients of variation are nearly identical across policies, and further in Figure 2, which shows that return-window violations were rare. Together these results indicate that modest, budgeted excursions did not erode regularity. Second, *operational* gating: Slack-Aware's headway-risk screen, which combines lateness and spacing information, accepted a side pickup only when the risk score was below its threshold. As shown in Table 2 and in Figures 1(a)–(b), this gate was overly conservative under

our parameterization: it rejected many otherwise feasible pickups, producing sizable increases in waiting times and abandonments relative to both Baseline and Myopic. Figure 2 also illustrates Slack-Aware's near-perfect compliance, but this was achieved precisely because so few excursions were accepted in the first place. The implication is that gating should be *tuned* rather than binary, and, where possible, made *soft* (e.g., dynamic thresholds that relax under low load or per-trip quotas that cap risk exposure) instead of hard exclusions.

## 4.3 Heterogeneity across geometry, demand, and variability

Benefits were concentrated in scenarios where geometry expanded the set of side stops reachable within the budget/window. This is visible in Figures 1(a)–(b), where scenarios with shorter side detours show larger gaps between Myopic and Baseline in waiting times and abandonments. For example, the mean wait fell from about 179 s under Baseline to 139 s under Myopic (Table 2), but in some scenarios with tight windows or long side costs the difference nearly vanished, as reflected in near-overlapping dots in the same panels. Higher stochastic variability in departures and link times narrowed the feasible window for detours and amplified the probability of overrunning the return window. This pattern is reflected in Figure 2, where Myopic shows the highest missed return-window rate (about 43%), Slack-Aware shows a lower but still notable rate (about 27%), and Baseline records none, illustrating the trade-off between accepting more excursions and maintaining strict compliance. When combined with conservative gating, this produced especially high abandonments under Slack-Aware, as seen in the upper cluster of points in Figures 1(a)–(b). These interactions underscore that the same policy can deliver different outcomes depending on corridor shape, side-stop placement, and demand volatility, as also anticipated in the factor settings summarized in Table 1.

## 4.4 Myopic vs. Slack-Aware: choosing between policies

*Myopic-Feasible* is preferable when the objective is to expand access at side stops during off-peak periods and corridor headways are already reasonably stable. As shown in Table 2, it yielded shorter waits (about 139 s vs. 179 s under Baseline) and fewer abandonments (12% vs. 20%) at essentially unchanged headway CVs (Figure 1(c)) and negligible excursion mileage (Figure 1(d)). *Slack-Aware* is attractive when strict headway regularity or punctuality must be guaranteed under higher variability *and* when its risk threshold is calibrated to avoid starving side pickups. In our experiments, however, the default threshold led to pronounced service losses: Figure 1(a)–(b) shows Slack-Aware consistently above both Baseline and Myopic in waits and abandonments, while Figure 2 confirms that its lower violation rate (27% vs. 43% for Myopic) was achieved by rejecting many feasible excursions. This imbalance points to the need for data-driven calibration. A practical rule is to start from Myopic-Feasible and add lightweight safeguards only where measured headway risk is demonstrably high.

## 4.5 Implications for practice

For agencies considering off-peak pilots on overlapping lines, a staged approach is advisable. (i) Begin with small, instrumented excursions (moderate budget and a relaxed-but-finite return window), logging rejoin compliance and headway deviations at control points. (ii) Default to Myopic-Feasible to harvest near-corridor opportunities; monitor headway CVs and punctuality by time-of-day and direction. (iii) Introduce targeted safeguards where data show risk hotspots: dynamic risk thresholds in high-burstiness intervals, per-trip caps on excursions, or brief control-point holds to absorb rejoin noise. Because excursion shares of bus-km are tiny in our experiments, off-corridor sharing can deliver *micro-access* gains at side stops with minimal operating cost, provided rejoin windows are enforced. Equity requires tracking outcomes by stop type: Myopic-Feasible tends to improve access at side stops with little corridor penalty, whereas aggressive gating protects the corridor but can systematically under-serve side stops; minimum-service guarantees for side stops and quota-based safeguards can mitigate these disparities. The policy levers studied here (budgets, windows, and control-point checks) align with tools already available in many operations control systems, simplifying translation to field pilots.

**Limitations.** The results reflect stylized off-peak operations on two partially overlapping lines with simplified demand and dwell models. Although the mechanisms—detour budgets, rejoin windows, and headway gating—are generic, effect sizes will depend on local geometry, dispatch processes, and

passenger patience distributions. The open simulator and datasets are intended for corridor-specific calibration before deployment.

## 5 Conclusion

This study addressed whether fixed-route buses on partially overlapping corridors can, during off-peak periods, improve passenger service by allowing *bounded off-corridor excursions* to nearby side stops, and how simple online accept/reject policies and safeguards should be designed to contain reliability risks. Our objective was to quantify the service–reliability trade-offs and provide actionable guidance on excursion budgets, return-to-corridor windows, and headway-aware gating.

Across a compact yet informative set of scenarios, permitting brief, budgeted excursions consistently expanded access to side stops and reduced unmet demand while leaving corridor regularity essentially unchanged. In particular, a lightweight *Myopic-Feasible* policy harvested near-corridor opportunities—lowering waits and abandonments—without detectable headway deterioration and with negligible excursion mileage. A stricter *Slack-Aware* gate protected regularity but required careful threshold calibration; when set too conservatively, it starved otherwise feasible side pickups. Taken together, these findings indicate that bounded excursions can enhance accessibility with minimal reliability penalties, provided return windows and risk thresholds are tuned to local conditions.

**Practical implications.** The results suggest a staged deployment: start with moderate time/link budgets and enforce finite rejoin windows; monitor headway variation at control points; and introduce targeted safeguards (e.g., dynamic thresholds or per-trip quotas) where measured risk is high. Because the operational footprint of excursions is small, agencies can pilot these mechanisms with modest instrumentation and clear compliance logging.

**Limitations.** The analysis relies on a stylized, discrete-event simulation of two lines under off-peak conditions with simplified demand, dwell, and running-time models. Effect sizes and thresholds will vary with corridor geometry, dispatch processes, and rider patience in real systems. The findings should therefore be interpreted as design guidance to be calibrated, not as universal prescriptions.

**Future work.** Promising directions include: (i) extending to multi-line networks with interacting overlaps; (ii) integrating with demand-responsive and other flexible services to coordinate shared coverage; (iii) systematic sensitivity analyses for budgets, windows, and quota rules; (iv) learning-based or adaptive risk thresholds that adjust to load and variability in real time; and (v) field piloting with empirical traces to calibrate models and validate outcomes against observed operations.

**Broader Impacts.** This study has potential positive impacts by expanding accessibility and equity in fixed-route transit through lightweight, reproducible mechanisms that agencies could adopt without major institutional change. At the same time, if safeguards such as detour budgets, rejoin windows, or risk thresholds are miscalibrated, service to side stops could become inequitable or corridor reliability could be degraded. These risks highlight the importance of calibration with local data and careful monitoring in pilot deployments. More broadly, our open simulator and datasets can help agencies and researchers explore these trade-offs transparently before real-world implementation.

**AI Agent Setup.** This paper was produced primarily by an AI agent, *AI Scientist (GPT-5)*, instantiated as the GPT-5 Thinking large language model. The agent operated in a tool-augmented single-agent loop that cycled between (i) planning and code generation, (ii) execution, testing, and error repair, and (iii) analysis and writing. The customized orchestration bound the model to a Python execution environment for developing and running the discrete-event simulator, performing data analysis, and generating figures/tables; provided file I/O over CSV/YAML/JSON for scenario manifests and run logs; and supported direct LaTeX editing/compilation for the manuscript. The simulator, policy implementations (baseline, myopic-feasible, slack-aware), experiment harness, and KPI aggregation were written by the agent in Python (NumPy, pandas, matplotlib, PyYAML), with deterministic seeds, per-run metadata, and reproducible outputs. No external web tools or retrieval systems were used for experiments; literature PDFs were supplied offline. Human collaborators offered light scoping guidance (e.g., factor reduction), uploaded PDFs for citation access, and made minor LaTeX adjustments; they did not author code or modify results.

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

# Agents4Science AI Involvement Checklist

1. **Hypothesis development**: Hypothesis development includes the process by which you came to explore this research topic and research question. This can involve the background research performed by either researchers or by AI. This can also involve whether the idea was proposed by researchers or by AI.

   Answer: [D]

   Explanation: The hypothesis was primarily developed by AI. The researchers contributed the initial interest in off-corridor service but relied on AI to identify and articulate the research gap, propose the hypothesis, and draft the initial paper proposal. The AI also provided the relevant literature; the researchers only downloaded the suggested papers from databases and re-uploaded them so the AI could process the full content. This makes hypothesis development largely AI-led, with researchers mainly facilitating access and validation.

2. **Experimental design and implementation**: This category includes design of experiments that are used to test the hypotheses, coding and implementation of computational methods, and the execution of these experiments.

   Answer: [D]

   Explanation: The experimental design, coding of the simulator, implementation of policies, and execution of runs were carried out by AI. The researchers' role was limited to guiding scope—specifically, suggesting a reduction in the number of experimental factors to keep the design tractable. Thus, this component was primarily AI-led, with minimal human input.

3. **Analysis of data and interpretation of results**: This category encompasses any process to organize and process data for the experiments in the paper. It also includes interpretations of the results of the study.

   Answer: [D]

   Explanation: Data processing, KPI computation, and interpretation of simulation outputs were conducted by AI, including generation of tables, figures, and narrative insights. The researchers' role was limited to suggesting more granular result displays in order to capture differences between scenarios. Overall, the analysis and interpretation were AI-led, with minimal human contribution.

4. **Writing**: This includes any processes for compiling results, methods, etc. into the final paper form. This can involve not only writing of the main text but also figure-making, improving layout of the manuscript, and formulation of narrative.

   Answer: [D]

   Explanation: The main text, figures, and narrative flow were written and compiled by AI. Researchers' contributions were limited to minor LaTeX adjustments (e.g., figure/table positioning, subfigure formatting) and light formatting edits. Thus, the writing process was overwhelmingly AI-led, with only superficial human input on manuscript layout.

5. **Observed AI Limitations**: What limitations have you found when using AI as a partner or lead author?

   Description: Two main limitations emerged. First, AI agents tend to push forward relentlessly, elaborating on any assumption or hypothesis regardless of its correctness or relevance. If a premise is flawed or only tangential, the AI will nonetheless develop it into full sections of a paper. This requires continuous human oversight to steer direction, correct errors, and suppress unproductive tangents—akin to taming a wild horse. Second, AI cannot access paywalled or restricted research articles directly. Researchers had to manually retrieve and upload the suggested literature so the AI could process it, underscoring the dependency on human mediation for data access.

